# Augmented curation of clinical notes from a massive EHR system reveals symptoms of impending COVID-19 diagnosis

Tyler Wagner[1†], FNU Shweta[2†], Karthik Murugadoss[1], Samir Awasthi[1], AJ Venkatakrishnan[1], Sairam Bade[3], Arjun Puranik[1], Martin Kang[1], Brian W Pickering[2], John C O'Horo[2], Philippe R Bauer[2], Raymund R Razonable[2], Paschalis Vergidis[2], Zelalem Temesgen[2], Stacey Rizza[2], Maryam Mahmood[2], Walter R Wilson[2], Douglas Challener[2], Praveen Anand[3], Matt Liebers[1], Zainab Doctor[1], Eli Silvert[1], Hugo Solomon[1], Akash Anand[3], Rakesh Barve[3], Gregory Gores[2], Amy W Williams[2], William G Morice II[2,4], John Halamka[2], Andrew Badley[2*], Venky Soundararajan[1*]

[1]nference, Cambridge, United States; [2]Mayo Clinic, Rochester, United States; [3]nference Labs, Bangalore, India; [4]Mayo Clinic Laboratories, Rochester, United States

**\*For correspondence:**
Badley.Andrew@mayo.edu (AB);
venky@nference.net (VS)

[†]These authors contributed equally to this work

**Abstract** Understanding temporal dynamics of COVID-19 symptoms could provide fine-grained resolution to guide clinical decision-making. Here, we use deep neural networks over an institution-wide platform for the augmented curation of clinical notes from 77,167 patients subjected to COVID-19 PCR testing. By contrasting Electronic Health Record (EHR)-derived symptoms of COVID-19-positive ($COVID_{pos}$; n = 2,317) versus COVID-19-negative ($COVID_{neg}$; n = 74,850) patients for the week preceding the PCR testing date, we identify anosmia/dysgeusia (27.1-fold), fever/chills (2.6-fold), respiratory difficulty (2.2-fold), cough (2.2-fold), myalgia/arthralgia (2-fold), and diarrhea (1.4-fold) as significantly amplified in $COVID_{pos}$ over $COVID_{neg}$ patients. The combination of cough and fever/chills has 4.2-fold amplification in $COVID_{pos}$ patients during the week prior to PCR testing, in addition to anosmia/dysgeusia, constitutes the earliest EHR-derived signature of COVID-19. This study introduces an *Augmented Intelligence* platform for the real-time synthesis of institutional biomedical knowledge. The platform holds tremendous potential for scaling up curation throughput, thus enabling EHR-powered early disease diagnosis.

## Introduction

As of June 3, 2020, according to WHO there have been more than 6.3 million confirmed cases worldwide and more than 379,941 deaths attributable to COVID-19 (https://covid19.who.int/). The clinical course and prognosis of patients with COVID-19 varies substantially, even among patients with similar age and comorbidities. Following exposure and initial infection with SARS-CoV-2, likely through the upper respiratory tract, patients can remain asymptomatic with active viral replication for days before symptoms manifest (*Guan et al., 2020*; *Gandhi et al., 2020*; *Verity et al., 2020*). The asymptomatic nature of initial SARS-CoV-2 infection patients may be exacerbating the rampant community transmission observed (*Hoehl et al., 2020*). It remains unknown why certain patients become symptomatic, and in those that do, the timeline of symptoms remains poorly characterized and non-specific. Symptoms may include fever, fatigue, myalgias, loss of appetite, loss of smell (anosmia), and altered sense of taste, in addition to the respiratory symptoms of dry cough, dyspnea, sore throat, and rhinorrhea, as well as gastrointestinal symptoms of diarrhea, nausea, and abdominal discomfort (*Xiao et al., 2020*). A small proportion of COVID-19 patients progress to

severe illness, requiring hospitalization or intensive care management; among these individuals, mortality due to Acute Respiratory Distress Syndrome (ARDS) is higher (*Zhang et al., 2020*). The estimated average time from symptom onset to resolution can range from three days to more than three weeks, with a high degree of variability (*Bi et al., 2020*). The COVID-19 public health crisis demands a data science-driven approach to quantify the temporal dynamics of COVID-19 pathophysiology. However, for this there is a need to overcome challenges associated with manual curation of unstructured EHRs in a clinical context (*Argenziano et al., 2020*) and self-reporting outside of the clinical settings via questionnaires (*Menni et al., 2020*).

Here we introduce a platform for the augmented curation of the full-spectrum of patient symptoms from the Mayo Clinic EHRs for 77,167 patients with positive/negative COVID-19 diagnosis by PCR testing (see Materials and methods). The platform utilizes state-of-the-art transformer neural networks on the unstructured clinical notes to automate entity recognition (e.g. diseases, symptoms), quantify the strength of contextual associations between entities, and characterize the nature of association into 'positive', 'negative', 'suspected', or 'other' sentiments. We identify specific sensory, respiratory, and gastro-intestinal symptoms, as well as some specific combinations, that appear to be indicative of impending $COVID_{pos}$ diagnosis by PCR testing. This highlights the potential for neural network-powered EHR curation to facilitate a significantly earlier diagnosis of COVID-19 than currently feasible.

## Results

The clinical determination of the COVID-19 status for each patient was conducted using the SARS-CoV-2 PCR (RNA) test approved for human nasopharyngeal and oropharyngeal swab specimens under the U.S. FDA emergency use authorization (EUA) (*Mayo Clinic Laboratories, 2019*). This PCR test resulted in 74,850 $COVID_{neg}$ patient diagnoses and 2,317 $COVID_{pos}$ patient diagnoses. The $COVID_{pos}$ cohort had a mean age of 41.9 years (standard deviation = 19.1 years) and was 51% male and 49% female while the $COVID_{neg}$ cohort had a mean age of 50.7 years (standard deviation = 21.4 years) and was 43% male and 57% female. Only 11 (0.5%) of $COVID_{pos}$ and 196 (0.3%) of $COVID_{neg}$ patients were hospitalized 7 days or more prior to PCR testing, indicating that the vast majority of patients were not experiencing serious illness prior to this time window. During the week prior to PCR testing, 135 (5.8%) of the $COVID_{pos}$ and 5981 (8.0%) of the $COVID_{neg}$ patients were hospitalized. Additionally, the frequencies of ICD10 diagnosis codes for these cohorts were found for the week prior to PCR testing, with unspecified acute upper respiratory infection appearing in over 20% of both cohorts (*Supplementary file 1a-b*). In order to investigate the time course of COVID-19 progression in patients and better define the presence or absence of symptoms, we used BERT-based deep neural networks to extract symptoms and their putative synonyms from the clinical notes for the week prior to the date when the COVID-19 diagnosis test was taken (see Materials and methods; *Table 1*). For the purpose of this analysis, all patients were temporally aligned, by setting the date of COVID-19 PCR testing to 'day 0', and the proportion of patients demonstrating each symptom derived from the EHR over each day of the week preceding testing was tabulated (*Table 2*). As a negative control, we included a non-COVID-19 symptom 'dysuria'.

Altered or diminished sense of taste or smell (dysgeusia or anosmia) is the most significantly amplified signal in $COVID_{pos}$ over $COVID_{neg}$ patients in the week preceding PCR testing (*Table 1*; 27.1-fold amplification; p-value<<1E-100). This result suggests that anosmia and dysgeusia are likely the most salient early indicators of COVID-19 infection, including in otherwise asymptomatic patients. However, it must be noted that the prevalence of a symptom in the population must be taken into consideration. Thus, while anosmia/ dysgeusia see the most dramatic difference in prevalence between the $COVID_{pos}$ and $COVID_{neg}$ cohorts, the overall prevalence of 318 out of 77,167 patients, or 0.4%, precludes it from being a standalone predictor of infection. However, with the recent addition of anosmia and dysgeusia to the CDC guidelines (*Mayo Clinic Laboratories, 2019*) these symptoms will likely be reported more frequently as the pandemic progresses. Alternatively, fever/chills also have an increased signal in the $COVID_{pos}$ compared to the $COVID_{neg}$ cohort (2.6-fold amplification; p-value = 3.6E-169) while appearing in 10,171 out of 77,167 patients (13.2%).

Diarrhea is also significantly amplified in the $COVID_{pos}$ patients for the week preceding PCR testing (*Table 1*; 1.4-fold; p-value = 3.5E-06). Some of these not yet-diagnosed COVID-19 patients that experience diarrhea prior to PCR testing may be unintentionally shedding SARS-CoV-2 fecally

**Table 1.** Augmented curation of the unstructured clinical notes from the EHR reveals specific clinically confirmed phenotypes that are amplified in COVID$_{pos}$ patients over COVID$_{neg}$ patients in the week prior to the SARS-CoV-2 PCR testing date.

The key COVID$_{pos}$ amplified symptoms in the week preceding PCR testing (i.e. day = −7 to day = −1) are highlighted in gray (p-value<1E-10). The ratio of COVID$_{pos}$ to COVID$_{neg}$ proportions represents the fold change amplification of each phenotype in the COVID$_{pos}$ patient set (symptoms are sorted based on this column).

| Symptom (p-value<1E-10 in gray) | COVID+ Count (%) (N = 2317) | COVID- Count (%) (N = 74850) | (COVID+/COVID-) Relative Ratio | Relative ratio (95% CI) | 2-tailed p-value | BH-corrected p-value |
|---|---|---|---|---|---|---|
| Altered or diminished sense of taste or smell | 145 (6.3%) | 173 (0.2%) | 27.08 | (21.81, 33.62) | <1E-300 | <1E-300 |
| Fever/chills | 750 (32.4%) | 9421 (12.6%) | 2.57 | (2.42, 2.74) | 3.57E-169 | 4.64E-168 |
| Cough | 769 (33.2%) | 11083 (14.8%) | 2.24 | (2.11, 2.38) | 4.60E-129 | 3.99E-128 |
| Respiratory difficulty | 681 (29.4%) | 10082 (13.5%) | 2.18 | (2.04, 2.33) | 3.06E-105 | 1.99E-104 |
| Myalgia/Arthralgia | 288 (12.4%) | 4620 (6.2%) | 2.01 | (1.8, 2.25) | 5.35E-34 | 2.78E-33 |
| Rhinitis | 200 (8.6%) | 2947 (3.9%) | 2.19 | (1.92, 2.52) | 2.25E-29 | 9.75E-29 |
| Headache | 325 (14.0%) | 6124 (8.2%) | 1.71 | (1.55, 1.9) | 1.34E-23 | 4.98E-23 |
| Congestion | 228 (9.8%) | 4261 (5.7%) | 1.73 | (1.53, 1.96) | 4.45E-17 | 1.45E-16 |
| GI upset | 195 (8.4%) | 10670 (14.3%) | 0.59 | (0.52, 0.68) | 1.74E-15 | 5.03E-15 |
| Wheezing | 49 (2.1%) | 3765 (5.0%) | 0.42 | (0.32, 0.56) | 1.82E-10 | 4.73E-10 |
| Dermatitis | 26 (1.1%) | 2519 (3.4%) | 0.33 | (0.23, 0.5) | 2.60E-09 | 6.15E-09 |
| Generalized symptoms | 169 (7.3%) | 8129 (10.9%) | 0.67 | (0.58, 0.78) | 4.82E-08 | 1.04E-07 |
| Respiratory Failure | 73 (3.2%) | 1363 (1.8%) | 1.73 | (1.38, 2.19) | 3.09E-06 | 6.18E-06 |
| Diarrhea | 228 (9.8%) | 5452 (7.3%) | 1.35 | (1.19, 1.53) | 3.47E-06 | 6.44E-06 |
| Pharyngitis | 160 (6.9%) | 3635 (4.9%) | 1.42 | (1.22, 1.66) | 7.05E-06 | 1.22E-05 |
| Chest pain/pressure | 148 (6.4%) | 6122 (8.2%) | 0.78 | (0.67, 0.92) | 1.88E-03 | 3.06E-03 |
| Change in appetite/intake | 95 (4.1%) | 2271 (3.0%) | 1.35 | (1.11, 1.66) | 3.37E-03 | 5.15E-03 |
| Otitis | 13 (0.6%) | 874 (1.2%) | 0.48 | (0.29, 0.85) | 6.98E-03 | 1.01E-02 |
| Cardiac | 95 (4.1%) | 2443 (3.3%) | 1.26 | (1.03, 1.54) | 2.62E-02 | 3.59E-02 |
| Fatigue | 229 (9.9%) | 8268 (11.0%) | 0.89 | (0.79, 1.02) | 7.83E-02 | 1.02E-01 |
| Conjunctivitis | 9 (0.4%) | 167 (0.2%) | 1.74 | (0.95, 3.52) | 1.00E-01 | 1.24E-01 |
| Dry mouth | 5 (0.2%) | 316 (0.4%) | 0.51 | (0.24, 1.3) | 1.28E-01 | 1.51E-01 |
| Hemoptysis | 13 (0.6%) | 283 (0.4%) | 1.48 | (0.89, 2.65) | 1.61E-01 | 1.78E-01 |
| Dysuria | 16 (0.7%) | 732 (1.0%) | 0.71 | (0.45, 1.18) | 1.64E-01 | 1.78E-01 |
| Diaphoresis | 35 (1.5%) | 979 (1.3%) | 1.15 | (0.84, 1.63) | 3.99E-01 | 4.15E-01 |
| Neuro | 150 (6.5%) | 4952 (6.6%) | 0.98 | (0.84, 1.15) | 7.86E-01 | 7.86E-01 |

(*Xu et al., 2020*; *Wu, 2020*). Incidentally, epidemiological surveillance by waste water monitoring conducted recently in the state of Massachusetts observed SARS-CoV-2 RNA (*Xu et al., 2020 Wu, 2020*). The amplification of diarrhea in COVID$_{pos}$ over COVID$_{neg}$ patients for the week preceding PCR testing raises concern for other modes of viral transmission and highlights the importance of washing hands frequently in addition to wearing respiratory protection.

As may be expected, respiratory difficulty is enriched in the week prior to PCR testing in COVID$_{pos}$ over COVID$_{neg}$ patients (1.9-fold amplification; p-value = 1.1E-22; *Table 1*). Among other common phenotypes with significant enrichments in COVID$_{pos}$ over COVID$_{neg}$ patients, cough has a 2.2-fold amplification (p-value = 4.6E-129) and myalgia/arthralgia has a 2.0-fold amplification (p-value = 5.3E-34). Rhinitis is also a potential early signal of COVID$_{pos}$ patients that requires some consideration (2.2-fold amplification, p-value = 2.25E-29). Finally, dysuria was included as a negative

**Table 2.** Temporal analysis of the EHR clinical notes for the week preceding PCR testing (i.e. day −7 to day −1), leading up to the day of PCR testing (day 0) in COVID$_{pos}$ and COVID$_{neg}$ patients.

Temporal enrichment for each symptom is quantified using the ratio of COVID$_{pos}$ patient proportion over the COVID$_{neg}$ patient proportion for each day. The patient proportions in the rows labeled 'Positive' and 'Negative' represent the fraction of COVID$_{pos}$ (n = 2,317) and COVID$_{neg}$ (n = 74,850) patients with the specified symptom on each day. Symptoms with p-value<1E-10 are highlighted in green and 1E-10 < p value<1E-03 in gray.

| Symptom | COVID-19 (N = 77167) | Day = −7 | Day = −6 | Day = −5 | Day = −4 | Day = −3 | Day = −2 | Day = −1 |
|---|---|---|---|---|---|---|---|---|
| Altered or diminished sense of taste or smell | Positive (n = 2317) | 4.75E-03 | 3.88E-03 | 3.45E-03 | 2.59E-03 | 1.73E-03 | 0.00E+00 | 4.75E-03 |
| | Negative (n = 74850) | 1.07E-04 | 4.01E-05 | 1.07E-04 | 1.07E-04 | 9.35E-05 | 2.27E-04 | 9.75E-04 |
| | Ratio (Positive/Negative) | 44.42 | 96.91 | 32.30 | 24.23 | 18.46 | 0.00 | 4.87 |
| | p-value | 1.14E-44 | 2.24E-48 | 3.17E-28 | 2.35E-18 | 8.94E-11 | 4.68E-01 | 5.85E-08 |
| Cough | Positive | 2.55E-02 | 2.29E-02 | 1.90E-02 | 1.64E-02 | 1.38E-02 | 8.63E-03 | 7.94E-02 |
| | Negative | 4.88E-03 | 5.30E-03 | 5.21E-03 | 5.33E-03 | 5.73E-03 | 8.40E-03 | 8.71E-02 |
| | Ratio (Positive/Negative) | 5.22 | 4.31 | 3.64 | 3.08 | 2.41 | 1.03 | 0.91 |
| | p-value | 8.42E-40 | 7.44E-28 | 2.43E-18 | 2.68E-12 | 6.68E-07 | 9.06E-01 | 1.95E-01 |
| Diarrhea | Positive | 8.20E-03 | 7.77E-03 | 6.04E-03 | 4.32E-03 | 4.75E-03 | 2.59E-03 | 2.68E-02 |
| | Negative | 3.70E-03 | 4.26E-03 | 4.58E-03 | 4.09E-03 | 4.58E-03 | 5.61E-03 | 3.78E-02 |
| | Ratio (Positive/Negative) | 2.22 | 1.82 | 1.32 | 1.06 | 1.04 | 0.46 | 0.71 |
| | p-value | 5.59E-04 | 1.17E-02 | 3.08E-01 | 8.66E-01 | 9.08E-01 | 5.32E-02 | 5.81E-03 |
| Fever/chills | Positive | 2.42E-02 | 2.20E-02 | 1.94E-02 | 1.68E-02 | 1.34E-02 | 6.47E-03 | 7.90E-02 |
| | Negative | 3.39E-03 | 3.74E-03 | 3.90E-03 | 4.42E-03 | 4.61E-03 | 6.77E-03 | 7.48E-02 |
| | Ratio (Positive/Negative) | 7.12 | 5.88 | 4.98 | 3.81 | 2.90 | 0.96 | 1.06 |
| | p-value | 1.15E-54 | 4.31E-40 | 6.52E-29 | 1.64E-17 | 2.36E-09 | 8.62E-01 | 4.52E-01 |
| Respiratory Difficulty | Positive | 2.24E-02 | 2.11E-02 | 1.81E-02 | 1.55E-02 | 1.25E-02 | 8.20E-03 | 5.35E-02 |
| | Negative | 5.06E-03 | 5.70E-03 | 5.81E-03 | 5.87E-03 | 6.16E-03 | 8.66E-03 | 7.65E-02 |
| | Ratio (Positive/Negative) | 4.43 | 3.71 | 3.12 | 2.65 | 2.03 | 0.95 | 0.70 |
| | p-value | 2.07E-28 | 8.72E-21 | 9.41E-14 | 4.56E-09 | 1.48E-04 | 8.15E-01 | 3.89E-05 |
| Change in appetite/intake | Positive | 1.73E-03 | 1.73E-03 | 1.73E-03 | 5.18E-03 | 4.32E-03 | 5.61E-03 | 1.86E-02 |
| | Negative | 1.30E-03 | 1.36E-03 | 1.34E-03 | 1.39E-03 | 1.40E-03 | 1.91E-03 | 1.35E-02 |
| | Ratio (Positive/Negative) | 1.33 | 1.27 | 1.29 | 3.73 | 3.08 | 2.94 | 1.37 |
| | p-value | 5.72E-01 | 6.42E-01 | 6.14E-01 | 3.53E-06 | 3.43E-04 | 9.41E-05 | 4.03E-02 |
| Myalgia/Arthralgia | Positive | 8.20E-03 | 9.06E-03 | 7.77E-03 | 6.47E-03 | 5.61E-03 | 2.59E-03 | 3.84E-02 |
| | Negative | 2.24E-03 | 3.05E-03 | 3.17E-03 | 2.99E-03 | 2.87E-03 | 3.99E-03 | 2.72E-02 |
| | Ratio (Positive/Negative) | 3.65 | 2.98 | 2.45 | 2.16 | 1.95 | 0.65 | 1.41 |
| | p-value | 9.33E-09 | 4.91E-07 | 1.44E-04 | 2.98E-03 | 1.68E-02 | 2.88E-01 | 1.18E-03 |
| Congestion | Positive | 6.91E-03 | 6.47E-03 | 5.18E-03 | 3.45E-03 | 5.18E-03 | 2.16E-03 | 1.94E-02 |
| | Negative | 1.95E-03 | 2.38E-03 | 1.98E-03 | 2.36E-03 | 2.18E-03 | 2.95E-03 | 2.63E-02 |
| | Ratio (Positive/Negative) | 3.54 | 2.72 | 2.62 | 1.46 | 2.38 | 0.73 | 0.74 |
| | p-value | 2.87E-07 | 1.01E-04 | 8.47E-04 | 2.92E-01 | 2.78E-03 | 4.86E-01 | 4.07E-02 |

*Table 2 continued on next page*

*Table 2 continued*

| Symptom | COVID-19 (N = 77167) | Day = −7 | Day = −6 | Day = −5 | Day = −4 | Day = −3 | Day = −2 | Day = −1 |
|---|---|---|---|---|---|---|---|---|
| Rhinitis | Positive | 7.77E-03 | 6.04E-03 | 4.32E-03 | 3.02E-03 | 2.16E-03 | 8.63E-04 | 1.38E-02 |
| | Negative | 1.23E-03 | 1.42E-03 | 1.32E-03 | 1.36E-03 | 1.38E-03 | 2.04E-03 | 1.96E-02 |
| | Ratio (Positive/Negative) | 6.32 | 4.27 | 3.26 | 2.22 | 1.57 | 0.42 | 0.70 |
| | p-value | 2.08E-16 | 2.61E-08 | 1.58E-04 | 3.63E-02 | 3.21E-01 | 2.11E-01 | 4.59E-02 |

control for COVID-19, and consistent with this assumption, 0.69% of $COVID_{pos}$ patients and 0.97% of $COVID_{neg}$ patients had dysuria during the week preceding PCR testing.

Next, we considered the 325 possible pairwise combinations of 26 symptoms (*Supplementary file 1c*) for $COVID_{pos}$ versus $COVID_{neg}$ patients in the week prior to the PCR testing date (*Supplementary file 1d*). As expected from the previous results, altered sense of smell or taste (anosmia/dysgeusia) dominates in combination with many of the aforementioned symptoms as the most significant combinatorial signature of impending $COVID_{pos}$ diagnosis (particularly along with cough, respiratory difficulty, and fever/chills). Examining the other 300 possible pairwise symptom combinations, excluding the altered sense of smell of taste, reveals other interesting combinatorial signals. The combination of cough and diarrhea is noted to be significant in $COVID_{pos}$ over $COVID_{neg}$ patients during the week preceding PCR testing; that is cough and diarrhea co-occur in 8.0% of $COVID_{pos}$ patients and only 2.8% of $COVID_{neg}$ patients, indicating a 2.8-fold amplification of this specific symptom combination (BH corrected p-value = 5.6E-32, *Supplementary file 1d*).

We further investigated the temporal evolution of the proportion of patients with each symptom for the week prior to PCR testing (*Table 2*). Altered sense of taste or smell, cough, diarrhea, fever/chills, and respiratory difficulty were found to be significant discriminators of $COVID_{pos}$ from $COVID_{neg}$ patients between 4 to 7 days prior to PCR testing. During that time period, cough is significantly amplified (>3 fold, p-value<0.05) in the $COVID_{pos}$ patient cohort over the $COVID_{neg}$ patient cohort by 5.2-fold on day −7 (p-value = 8.4E-40), 4.31-fold on day −6(p-value = 7.44E-28), 3.6-fold on day −5 (p-value = 2.4E-18), and 3.1-fold on day −4 (p-value = 2.7E-12). The diminishing odds of cough as a symptom from 7 to 4 days preceding the PCR testing date is notable and this temporal pattern could potentially suggest that the duration of cough and other symptoms, in addition to their presence or absence, is a useful indicator of infection. Similarly, diarrhea is amplified in the $COVID_{pos}$ patient cohort over the $COVID_{neg}$ patient cohort for days furthest preceding from the PCR testing date, with an amplification of 2.2-fold on day −7 (p-value = 5.6E-04) and 1.8-fold on day −6 (p-value = 1.2E-02). Likewise, fever/chills and respiratory difficulty both show matching trends, with significant amplification in the $COVID_{pos}$ cohort on days −7 to −4 and days −7 to −5, respectively. However, unlike diarrhea, cough, fever/chills, and respiratory difficulty, we find that change in appetite may be considered a subsequent symptom of impending COVID-19 diagnosis, with significant amplification in the $COVID_{pos}$ cohort over the $COVID_{neg}$ cohort on day −4 (3.7-fold, p-value = 3.53E-06), day −3 (3.1-fold, p-value = 3.4E-04), and day −2 (2.9-fold, p-value = 9.4E-05). The delay in the onset of change in appetite/intake compared to the other aforementioned symptoms indicates that this change only appears after other symptoms have already manifested and thus could be secondary to these symptoms rather than directly caused by infection.

This high-resolution temporal overview of the EHR-derived clinical symptoms as they manifest prior to the SARS-CoV-2 PCR diagnostic testing date for 77,167 patients has revealed specific enriched signals of impending COVID-19 onset. These clinical insights can help modulate social distancing measures and appropriate clinical care for individuals exhibiting the specific sensory (anosmia, dysgeusia), respiratory (cough, difficulty breathing), gastro-intestinal (diarrhea, change in appetite/intake), and other (fever/chills, arthralgia/myalgia) symptoms identified herein, including for patients awaiting conclusive COVID-19 diagnostic testing results (e.g. by SARS-CoV-2 RNA RT-PCR).

## Discussion

While PCR testing is the current diagnostic standard of COVID-19, identifying risk of a positive diagnosis earlier is essential to mitigate the spread of the virus. Patients with these symptom risk factors could be tested earlier, undergo closer monitoring, and be adequately quarantined to not only ensure better treatment for the patient, but to prevent the infection of others. Additionally, as businesses begin to reopen, understanding these risk factors will be critical in areas where comprehensive PCR testing is not possible. This study demonstrates how such symptoms can be extracted from highly unstructured institutional knowledge and synthesized using deep learning and neural networks (*Devlin et al., 2019*). Such augmented curation, providing fine-grained, temporal resolution of symptoms, can be applied toward supporting differential diagnosis of patients in a clinical setting. Expanding beyond one institution's COVID-19 diagnostic testing and clinical care to the EHR databases of other academic medical centers and health systems will provide a more holistic view of clinical symptoms enriched in $COVID_{pos}$ over $COVID_{neg}$ patients in the days preceding confirmed diagnostic testing. This requires leveraging a privacy-preserving, federated software architecture that enables each medical center to retain the span of control of their de-identified EHR databases, while enabling the machine learning models from partners to be deployed in their secure cloud infrastructure. To this end, seamless multi-institute collaborations over an Augmented Intelligence platform, which puts patient privacy and HIPAA-compliance first, are being advanced actively over the Mayo Clinic's Clinical Data Analytics Platform Initiative (CDAP). The capabilities demonstrated in this study for rapidly synthesizing unstructured clinical notes to develop an EHR-powered clinical diagnosis framework will be further strengthened through such a universal biomedical research platform.

There are a few caveats that must be considered when relying solely on EHR inference to track symptoms preceding the PCR testing date. In addition to concerns regarding testing accuracy, there is an inherent delay in PCR testing, which arises because both the patient and physician must decide the symptoms warrant PCR testing. More specifically, to be tested, the patient must first consider the symptoms serious enough to visit the clinic and then the physician must determine the symptoms present a possibility of COVID infection. The length of this delay could also be influenced by how well-informed the public is of COVID-19 signs and symptoms, the availability of PCR testing, and the hospital protocols used to determine which patients get tested. Each of these factors would be absent or limited at the beginning of a pandemic but would increase or improve over time. This makes synchronization across patients difficult because the delay between symptom onset and PCR testing changes over time. For example, patients infected early in the pandemic would be less inclined to visit the clinic with mild symptoms, while those infected later have more information and more cause to get tested earlier. Similarly, in the early stages of the COVID-19 pandemic when PCR testing was limited, physicians were forced to reserve tests for more severe cases or for those who were in direct contact with a $COVID_{pos}$ individual, whereas now PCR testing is more widespread. In each case, the delay between symptom onset and PCR testing would be expected to change over time for a given patient population.

Additionally, there are caveats surrounding data availability when working with such real-world datasets, including data sparsity and reporting. For example, while each patient had at least one clinic visit, accompanied by physician notes, between days −7 and 0, only 6 (0.3%) of the $COVID_{pos}$ and 372 (0.5%) of the $COVID_{neg}$ patients had notes from all 7 days prior to PCR testing. Moreover, the number of patients with notes prior to PCR testing tends to decrease with each day prior to the PCR testing date for both cohorts (*Supplementary file 1e*). With regard to reporting, mild symptoms, particularly those seemingly unrelated to presentation for clinical care, such as anosmia, may go unreported. Finally, it should also be noted that $COVID_{neg}$ patients may not necessarily be representative of a 'healthy' cohort comparable to a randomly selected group from the general population, as these patients each had a reason for seeking out COVID-19 PCR testing. With all of these caveats in mind, the fact that the temporal distribution of symptoms significantly differs between the $COVID_{pos}$ and $COVID_{neg}$ patients remains and demonstrates that synchronization using the PCR testing date is apt for the real-world data analysis described herein. Thus, by understanding the temporal progression of symptoms prior to PCR testing, we aim to reduce the delay between infection and testing in the future.

As we continue to understand the diversity of COVID-19 patient outcomes through holistic inference of EHR systems, it is equally important to invest in uncovering the molecular mechanisms

(*Anand et al., 2020*) and gain cellular/tissue-scale pathology insights through large-scale patient-derived biobanking and multi-omics sequencing (*Venkatakrishnan et al., 2020*). To correlate patterns of molecular expression with EHR-derived symptom signals of COVID-19 disease progression, a large-scale bio-banking system has to be created. Such a system will enable deep molecular insights into COVID-19 to be gleaned and triangulated with SARS-CoV-2 tropism and patient outcomes, allowing researchers to better evaluate disease staging and synchronize patients for analyses similar to those presented here. Ultimately, connecting the dots between the temporal dynamics of COVID$_{pos}$ and COVID$_{neg}$ clinical symptoms across diverse patient populations to the multi-omics signals from patient-derived bio-specimen will help advance a more holistic understanding of COVID-19 pathophysiology. This will set the stage for a precision medicine approach to the diagnostic and therapeutic management of COVID-19 patients.

## Materials and methods

### Augmented curation of EHR patient charts

The nferX Augmented Curation technology was leveraged to rapidly curate the charts of SARS-CoV-2-positive (COVID$_{pos}$) patients. First, we read through the charts of 100 COVID$_{pos}$ patients and identified symptoms, grouping them into sets of synonymous words and phrases. For example, 'SOB', 'shortness of breath', and 'dyspnea', among others, were grouped into 'shortness of breath'. For the SARS-CoV2-positive patients, we identified a total of 26 symptom categories (*Supplementary file 1c*) with 145 synonyms or synonymous phrases. Together, these synonyms and synonymous phrases capture how symptoms related to COVID-19 are described in the Mayo Clinic Electronic Health Record (EHR) databases.

Next, for charts that had not yet been manually curated, we used state-of-the-art BERT-based neural networks (*Devlin et al., 2019*) to classify symptoms as being present or not present in each patient based on the surrounding phraseology. More specifically, SciBERT (*Beltagy et al., 2019*), a BERT model pre-trained on 3.17B tokens from the biomedical and computer science domains, was compared to both domain-adapted BERT architectures (e.g. BioBERT [*Lee et al., 2019*], ClinicalBio-BERT [*Alsentzer et al., 2019*]) and different transformer architectures (e.g. XLNet [*Yang et al., 2019*], RoBERTa [*Liu et al., 2019a*], MT-DNN [*Liu et al., 2019b*]). We found that SciBERT performed equally or better than these models (*Supplementary file 1f* and data not shown). SciBERT differs from other domain-adapted BERT architectures as it is trained de novo on a biomedical corpus, whereas BioBERT is initialized with the BERT base vocabulary and fine-tuned with PubMed abstracts and PMC articles. Similarly, Clinical BioBERT is initialized with BioBERT and fine-tuned on MIMIC-III data. When comparing different transformer architectures, SciBERT and RoBERTa had equivalent performance, slightly better than the other models tested, including XLNet and MT-DNN (data not shown). Thus, SciBERT was chosen for the analyses performed, using the architecture and training configuration shown in *Figure 1* and *Figure 1—figure supplement 1*.

The neural network used to perform this classification was initially trained using 18,490 sentences containing nearly 250 different cardiovascular, pulmonary, and metabolic diseases and phenotypes. Each sentence was manually classified into one of four categories: 'Yes' (confirmed phenotype), 'No' (ruled out phenotype), 'Maybe' (suspected phenotype), and 'Other' (alternate context, e.g. family history of a phenotype, risk of adverse event from medication, etc.), with examples of each classification shown in *Figure 1—figure supplement 2*. Using a 90%:10% train:test split, the model achieved 93.6% overall accuracy and a precision and recall of 95% or better for both positive and negative sentiment classification (*Supplementary file 1g*). To augment this model with COVID-related symptoms, 3188 sentences containing 26 different symptoms were added to the 18,490 previously tagged sentences for a total of 21,678. Classification was performed using that same labels and model performance was equivalent to the previous model, with an overall accuracy of 94.0% and a precision and recall of 96% or better for both positive and negative sentiment classification (*Supplementary file 1h*).

This model was first applied to 35,790,640 clinical notes across the entire medical history of 2,317 COVID$_{pos}$ patients and 74,850 COVID$_{neg}$ patients. Each patient is counted only once. Once they have a positive SARS-COV-2 PCR test, they are considered COVID$_{pos}$. If a patient were to test negative and then positive subsequently, that day of positive PCR testing is considered day 0 for that

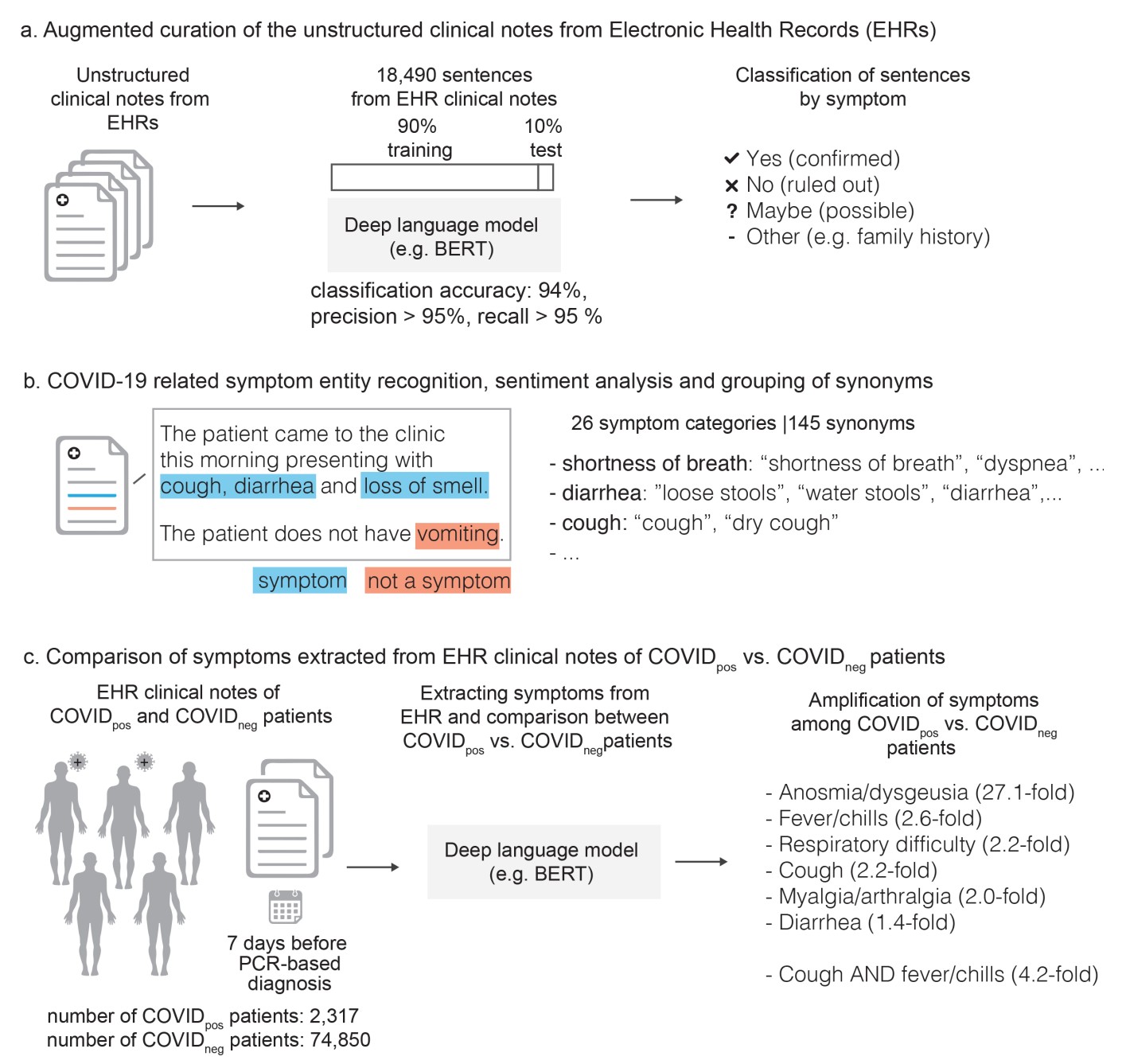

**Figure 1.** Augmented curation of the unstructured clinical notes and comparison of symptoms between COVIDpos vs. COVIDneg patients. (a) Augmented curation of the unstructured clinical notes from Electronic Health Records (EHRs). (b) COVID-19-related symptom entity recognition, sentiment analysis and grouping of synonyms. (c) Comparison of symptoms extracted from EHR clinical notes of COVID$_{pos}$ vs. COVID$_{neg}$ patients. The online version of this article includes the following figure supplement(s) for figure 1:

**Figure supplement 1.** SciBERT Architecture and Training Configuration.

**Figure supplement 2.** Examples of Sentence Classification Used in Training a SciBERT Model for Phenotype/Symptom Sentiment Analysis.

patient. We then focus on the notes from seven days prior to the SARS-CoV-2 diagnostic test (*Supplementary file 1e*). For each patient, the difference between the date on which a particular note was written and the PCR testing date were used to compute the relative date for that note. The PCR testing date was treated as 'day 0' with notes preceding it assigned 'day −1', 'day −2',

and so on. BERT-based neural networks were applied on each note to identify the set of symptoms that were present at that time for each patient. This patient-to-symptom mapping over time was then inverted to determine the set of unique patients experiencing each symptom at any given time. Here, the presence of a symptom was defined as either a 'Yes' or 'Maybe' classification by the model. The 'Maybe' classification was included because of differences in how phenotypes/diseases and symptoms are described in the clinical notes. For example, when physicians describe 'evaluation for' a phenotype/disease, for example 'the patient underwent evaluation for COVID-19', it does not imply a diagnosis for the disease, rather the possibility of a diagnosis. On the other hand, when a patient is seen 'for evaluation of cough, fever, and chills', this statement suggests that these symptoms are present. Thus, we included both classifications for the definition of a symptom being present.

To validate the accuracy of the BERT-based model for COVID-related symptoms, a validation step in which the classifications of 4001 such sentences from the timeframe of highest interest (day 0 to day −7) were manually verified. Sentences arising from templates, such as patient education documentation, accounted for 10.2% of sentences identified. These template sentences were excluded from the analysis. The true positive rate, defined as the total number of correct classifications divided by the number of total classifications, achieved by the model for classifying all symptoms was 96.7%; the corresponding false positive rate was 6.1%. The model achieved true positive rates ranging from 93% to 100% for the major symptom categories of Fever/Chills, Cough, Respiratory Difficulty, Headache, Fatigue, Myalgia/Arthralgia, Dysuria, Change in appetite/intake, and Diaphoresis. Classification performance was slightly lower for Altered or diminished sense of taste and smell; here, the true positive rate was 82.2%. Detailed statistics are displayed in *Supplementary file 1i*.

For each synonymous group of symptoms, we computed the count and proportion of $COVID_{pos}$ and $COVID_{neg}$ patients that had positive sentiment for that symptom in at least one note between 1 and 7 days prior to their PCR test. We additionally computed the ratio of those proportions to determine the prevalence of the symptom in the $COVID_{pos}$ cohort as compared to the $COVID_{neg}$ cohort; we then computed 95% confidence intervals around these ratios. A standard 2-proportion z hypothesis test was performed, and a p-value was reported for each symptom. A Benjamini-Hochberg adjustment was then applied on these 26-symptom p-values to account for multiple hypotheses. To capture the temporal evolution of symptoms in the $COVID_{pos}$ and $COVID_{neg}$ cohorts, the process was repeated considering counts and proportions for each day independently. (note we report unadjusted p-values only in the temporal analysis). Pairwise analysis of phenotypes was performed by considering 325 symptom pairs from the original set of 26 individual symptoms. For each pair, we calculated the number of patients in the $COVID_{pos}$ and $COVID_{neg}$ cohorts wherein both symptoms occurred at least once in the week preceding PCR testing. With these patient proportions, a Fisher exact test p-value was computed. A Benjamini-Hochberg adjustment was applied on these 325 Fisher test p-values to account for multiple hypothesis testing.

This research was conducted under IRB 20–003278, '*Study of COVID-19 patient characteristics with augmented curation of Electronic Health Records (EHR) to inform strategic and operational decisions*'. All analysis of EHRs was performed in the privacy-preserving environment secured and controlled by the Mayo Clinic. nference and the Mayo Clinic subscribe to the basic ethical principles underlying the conduct of research involving human subjects as set forth in the Belmont Report and strictly ensures compliance with the Common Rule in the Code of Federal Regulations (45 CFR 46) on Protection of Human Subjects.

## Acknowledgements

We thank Murali Aravamudan, Ajit Rajasekharan, Yanshan Wang, and Walter Kremers for their thoughtful review and feedback on this manuscript. We also thank Andrew Danielsen, Jason Ross, Jeff Anderson, Ahmed Hadad, and Sankar Ardhanari for their support that enabled the rapid completion of this study.

## Additional information

### Competing interests

Tyler Wagner, Karthik Murugadoss, Samir Awasthi, AJ Venkatakrishnan, Sairam Bade, Arjun Puranik, Martin Kang, Praveen Anand, Matt Liebers, Zainab Doctor, Eli Silvert, Hugo Solomon, Akash Anand, Rakesh Barve, Venky Soundararajan: is an employee of nference and has financial interests in the company. FNU Shweta, Brian W Pickering, John C O'Horo, Philippe R Bauer, Raymund R Razonable, Paschalis Vergidis, Zelalem Temesgen, Stacey Rizza, Maryam Mahmood, Walter R Wilson, Douglas Challener, Gregory Gores, Amy W Williams, William G Morice II, John Halamka, Andrew Badley: has a Financial Conflict of Interest in technology used in the research and with Mayo Clinic may stand to gain financially from the successful outcome of the research. This research has been reviewed by the Mayo Clinic Conflict of Interest Review Board and is being conducted in compliance with Mayo Clinic Conflict of Interest policies.

### Funding

| Funder | Grant reference number | Author |
| --- | --- | --- |
| National Institute of Allergy and Infectious Diseases | AI110173 | Andrew Badley |
| National Institute of Allergy and Infectious Diseases | AI120698 | Andrew Badley |

The funders had no role in study design, data collection and interpretation, or the decision to submit the work for publication.

### Author contributions

Tyler Wagner, Data curation, Formal analysis, Supervision, Validation, Investigation, Methodology, Writing - original draft, Writing - review and editing; FNU Shweta, Data curation, Formal analysis, Validation, Investigation, Writing - original draft, Project administration, Writing - review and editing; Karthik Murugadoss, Software, Formal analysis, Validation, Investigation, Methodology, Writing - original draft, Writing - review and editing; Samir Awasthi, Data curation, Formal analysis, Validation, Investigation, Methodology, Writing - original draft, Writing - review and editing; AJ Venkatakrishnan, Formal analysis, Validation, Investigation, Visualization, Methodology, Writing - original draft, Writing - review and editing; Sairam Bade, Software, Formal analysis, Investigation, Methodology; Arjun Puranik, Software, Formal analysis, Validation, Investigation, Visualization, Methodology; Martin Kang, Data curation, Validation, Investigation, Writing - original draft, Writing - review and editing; Brian W Pickering, Raymund R Razonable, Zelalem Temesgen, Stacey Rizza, Maryam Mahmood, Walter R Wilson, Douglas Challener, Gregory Gores, Amy W Williams, William G Morice II, John Halamka, Resources, Supervision, Validation, Investigation, Methodology, Project administration, Writing - review and editing; John C O'Horo, Resources, Validation, Investigation, Visualization, Methodology, Project administration, Writing - review and editing; Philippe R Bauer, Paschalis Vergidis, Resources, Supervision, Investigation, Methodology, Writing - original draft, Project administration, Writing - review and editing; Praveen Anand, Formal analysis, Investigation, Methodology, Writing - review and editing; Matt Liebers, Supervision, Visualization, Project administration; Zainab Doctor, Supervision, Investigation, Visualization, Project administration; Eli Silvert, Hugo Solomon, Visualization, Writing - original draft, Writing - review and editing; Akash Anand, Software, Formal analysis, Validation, Methodology; Rakesh Barve, Resources, Software, Formal analysis, Supervision, Investigation, Methodology, Project administration; Andrew Badley, Conceptualization, Resources, Formal analysis, Supervision, Funding acquisition, Validation, Investigation, Writing - original draft, Project administration, Writing - review and editing; Venky Soundararajan, Conceptualization, Resources, Formal analysis, Supervision, Funding acquisition, Validation, Investigation, Methodology, Writing - original draft, Project administration, Writing - review and editing

Author ORCIDs
FNU Shweta https://orcid.org/0000-0001-6634-6272
AJ Venkatakrishnan https://orcid.org/0000-0003-2819-3214
Douglas Challener http://orcid.org/0000-0002-6964-9639
Praveen Anand http://orcid.org/0000-0002-2478-7042
Venky Soundararajan https://orcid.org/0000-0001-7434-9211

### Ethics

Human subjects: This research was conducted under IRB 20-003278, "Study of COVID-19 patient characteristics with augmented curation of Electronic Health Records (EHR) to inform strategic and operational decisions". All analysis of EHRs was performed in the privacy-preserving environment secured and controlled by the Mayo Clinic. nference and the Mayo Clinic subscribe to the basic ethical principles underlying the conduct of research involving human subjects as set forth in the Belmont Report and strictly ensures compliance with the Common Rule in the Code of Federal Regulations (45 CFR 46) on Protection of Human Subjects. Please refer to the Mayo Clinic IRB website for further information - https://www.mayo.edu/research/institutional-review-board/overview.

### Decision letter and Author response

Decision letter https://doi.org/10.7554/eLife.58227.sa1
Author response https://doi.org/10.7554/eLife.58227.sa2

## Additional files

### Supplementary files

• Supplementary file 1. Enrichment of diagnosis codes. (**A**) Enrichment of diagnosis codes amongst COVID$_{pos}$ patients in the week preceding PCR testing. (**B**) Enrichment of diagnosis codes amongst COVID$_{neg}$ patients in the week preceding PCR testing. (**C**) Symptoms and their synonyms used for the EHR analysis. (**D**) Pairwise analysis of symptoms in the COVID$_{pos}$ and COVID$_{neg}$ cohorts. The pairwise symptom combinations with BH-corrected p-value<0.01 are summarized. (**E**) Patients with at least one clinical note over time. (**F**) SciBERT vs. BioClinicalBERT Phenotype Sentiment Model Performance on 18,490 sentences. (**G**) Model Performance Trained on 18,490 Sentences Containing 250 Different Cardiovascular, Pulmonary, and Metabolic Phenotype. (**H**) Model Performance Trained on 21,678 Sentences Containing 250 Different Cardiovascular, Pulmonary, and Metabolic Phenotypes and Expanded to Include 26 COVID-related Symptoms. (**I**) Synonym classification model performance

• Transparent reporting form

### Data availability

The EHR dataset where augmented curation was conducted from the Mayo Clinic records was accessed under IRB 20-003278, "Study of COVID-19 patient characteristics with augmented curation of Electronic Health Records (EHR) to inform strategic and operational decisions". The EHR data cannot be shared or released due to HIPAA regulations. Contact corresponding authors for additional details regarding the IRB, and please refer to the Mayo Clinic IRB website for further details on our commitment to patient privacy (https://www.mayo.edu/research/institutional-review-board/overview). The summary statistics derived from the EHRs are enclosed within the manuscript.

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
