## [Decision Letter]

**Acceptance summary:**

The concept of using EHR system and machine learning in COVID and finding anosmia as a specific clinical sign without a specific trigger for doctors to ask this sign is a very nice example of this system to help filter out new signatures in new or old diseases. The manuscript shows how to use big data to elucidate relevant clinical clues in practice.

**Decision letter after peer review:**

Thank you for submitting your article "Augmented Curation of Unstructured Notes from a Massive EHR System Reveals the Signature of Impending COVID-19 Diagnosis" for consideration by *eLife*. Your article has been reviewed by two peer reviewers, and the evaluation has been overseen by a Reviewing Editor and Jos van der Meer as the Senior Editor. The reviewers have opted to remain anonymous.

The reviewers have discussed the reviews with one another and the Reviewing Editor has drafted this decision to help you prepare a revised submission.

As the editors have judged that your manuscript is of interest, but as described below that it needs revision before it is published, we would like to draw your attention to changes in our revision policy that we have made in response to COVID-19 (https://elifesciences.org/articles/57162). We are offering, if you choose, to post the manuscript to bioRxiv (if it is not already there) along with this decision letter and a formal designation that the manuscript is "in revision at *eLife*". Please let us know if you would like to pursue this option. (If your work is more suitable for medRxiv, you will need to post the preprint yourself, as the mechanisms for us to do so are still in development.)

Summary:

The manuscript provides a deep neural networks study over an institution wide EHR platform for the augmented curation of 15.8 million clinical notes from 30,494 patients subjected to COVID-19 PCR testing during the SARS-CoV2 pandemic. The studies focused on potential earlier diagnosis of COVID-19 by identifying specific gastro-intestinal, respiratory, and sensory phenotypes, as well as some of their specific combinations. Overall, the research question is interesting, and would contribute to the understanding of COVID-19 early diagnosis. This part of the manuscript is strong and justifies its publication.

However, the manuscript overall needs some reorganization, as there are sections from the Materials and methods and Discussion that belong in the Results, and the overall details about the data, methodology and discussion need better clarity and expansion.

The data on RNAseq and treatment strategies (Figure 1 and 2) do not belong in the context of this paper.

Essential revisions:

1) The cohort itself is not well-defined and should be better described (see below).

2) Materials and methods section should be written more clearly (see below).

3) Remove RNA seq data and treatment figures and discussion about these topics. It is distractive and does not make the manuscript stronger.

Cohort description:

How many patients had an admission 7 days before getting a swab taken? How many individuals do the authors have a full week of information on prior to the PCR? Why was there a delay in PCR testing of these patients? Who are the 30,000 patients, and why were they admitted to a hospital? Or were they outpatients? The setting and patient population needs to be described. Completely basic information like sex and age are missing. The authors report that they use 15.8 million clinical notes from 30,494 patients, so about 500 notes/patient. Are these from the 7-day period, so more than 70 notes written per day? Or from a longer non-disclosed time-span?

Materials and methods section:

In this study patients are analyzed until date of diagnosis (test), it seems to be an analysis of when in the natural course one chooses to test? As it is formulated in the manuscript, "temporal patterns" first of all indicate, that the population converge towards day of test, so patients "progress towards same phenotype" and it is unclear how this relates to COVID-19 progression? What is the link between the analysis and the references to chloroquine/hydroxychloroquine?

The presentation in Materials and methods, also has many unclear aspects. For example, what was the output from also curating disease and medication? It seems, that only symptoms are presented in the manuscript? How do symptom categories and phenotypes differ? The iteration for optimization of the model seems a little unclear and how were the 18,500 sentences in the test set selected? What was the indication for COVID-testing in these patients? Were they all hospitalized for different conditions? Were all 30,494 under suspect for COVID-19 or were some tested simply because they were hospitalized in the period of the pandemic (i.e. routine/screening test)? And what were the diagnoses of the COVIDneg cases? Where there notes on all patients from index -7 to 0 as mentioned above? What are the demographics of these patients? And were symptoms handled as chronic or temporary conditions? Why was altered or diminished sense of taste and smell (anosmia/dysgeusia) included in Results despite a classification performance of 64.4%? Not sure why there are two F scores. Why calculate F scores on sentences labeled as "not present" – how is recall not undefined in such a calculation? How were the sentences in step 2 and 3 chosen? Why is the sum of the true positive rate and the false positive rate not 100%? Confidence intervals would help the interpretation of the data. It would be great if the authors would provide number of tests or the significance level to help interpret the Benjamini-Hochberg correction.

2) The authors state that the platform utilizes state-of-the-art transformer neural networks. But they used BERT (original version) indeed. Bert is not the state-of-the-art transformer model. XLNet and RoBERTa are the state-of-the-art models. For name entity recognition of clinical note data, there have been some specific BERT-based variations or pre-trained models, such as BioBERT.

The authors did not provide details of the implementation of BERT configuration, like how many layers, heads? How they train BERT, like how many epochs? what is the optimizer setting? Did they use a pre-train (on which corpus) BERT? etc. All these details need to be provided for reduplication. My suggestion is better to provide several examples regarding what the input looks like ,i.e., the unstructured clinical notes, and corresponding structured output, so that readers can understand how powerful the model is.

3) The dataset split (90%/10%) is not usual. "70%/10%/20%" or "60%/10%/30%" are more common for fair evaluation in deep learning (the middle one represents the validation set). Could the authors provide the reason why they split dataset in this way or provide a reference that applied this strategy?

4) In the Results section, the first paragraph is about details of data collection, it would be more appropriate in part of the Materials and methods, and the final paragraph would be more appropriate in the Discussion. In Table 1, the proportion columns would be better to combine with the patients-count columns.

5) In terms of the Discussion, it would be important to emphasize and discuss the main findings. "By contrasting EHR-derived phenotypes of COVID pos (n = 635) versus COVID neg (n = 29,859) patients during the week preceding PCR testing, we identify anosmia/dysgeusia (37.4-fold), myalgia/arthralgia (2.6-fold), diarrhea (2.2-fold), fever/chills (2.1-fold), respiratory difficulty (1.9-fold), and cough (1.8-fold) as significantly amplified in COVID pos patients" This statement in the Abstract should be the key finding, but the authors did not emphasize the statistical method and make discussion properly.

6) The pairwise analysis of phenotypes considered only the combination of 2 phenotypes. How to process the combination of multiple phenotypes?

7) As the NLP-based entity recognition can bring errors, the following statistic analysis could be biased by these errors. The authors should emphasize this point. I would suggest the authors to use the manually curated data from the first 100 patients to perform the same analysis to see if it can generate the same results.

[Editors' note: further revisions were suggested prior to acceptance, as described below.]

Thank you for resubmitting your article "Augmented Curation of Clinical Notes from a Massive EHR System Reveals Symptoms of Impending COVID-19 Diagnosis" for consideration by *eLife*. Your article has again been reviewed by two peer reviewers, and the evaluation has been overseen by a Reviewing Editor and Jos van der Meer as the Senior Editor. The reviewers have opted to remain anonymous.

You have definitely made a number of changes that have improved the readability and consistency of the manuscript. Although the paper essentially only presents already known symptoms of the disease (e.g https://doi.org/10.1136/bmj.m1996), it is now clearer that the paper aims to characterize when and how long prior to PCR-diagnosis a symptom is overrepresented.

However, one reviewer comments that there are still many inconsistencies in the manuscript that at best makes it difficult to read but also bring uncertainty about the validity of the results. Materials and methods section paragraphs are still hard to follow to the extent that the study would be hard to reproduce.

Specific comments:

1) Altered smell is as mentioned not a new finding, the authors also report this in relatively small numbers (6.3%) for covid19 patients, it seems unclear how much of an impact this would have clinically. The authors should comment on how this differs from other studies. In the context Tim Spector's work using app may also be relevant.

2) I could not follow the number of symptoms (26 or 27?) and how were these 26 or 27 selected? Who selected these and what was their backgroundsWhy not search for overrepresented symptoms in general and include way more symptoms? Currently only suspected/known symptoms were included. "Examining the other 325 possible pairwise symptom combinations" which were the first 26 for example. In Table 1, why is respiratory failure and dysuria marked in a light grey color?

3) Why is there automated entity recognition of "drugs"? Drug are not otherwise mentioned in the paper? There is still a part about scRNA-sequencing in the results, but no results about RNA-seq presented.

4) Why was the method applied to the 35 million notes? When only notes from 7 days are analyzed in the study… I don't agree with deleting information about the number of notes, it does not solve the problem from the original submission. It would be nice for the reader to be able to evaluate the number of notes or words written about the included patients. Also, how many notes are from day 0? Are notes recorded at day 0 included or not, there seems to be inconsistency here throughout the manuscript. The BERT validation seems to be done on notes including day 0, whereas the actual proportion analyses seem to be excluding day 0 which is the day one must expect to have most notes concerning covid19.

5) Many basic aspects are still unclear. How many data points could one patient contribute with? Could one patient tested multiple times contribute more than once? If a patient was tested negative and then positive, how was this handled?

6) It a considerable limitation that there is no separation between chronic and acute symptoms. If the condition is chronic it wouldn't help at all in the diagnosis, only that this population for some reason is more prone to be covid19 positive.

7) It still remains unclear why patients with obvious covid19 symptoms like "respiratory difficulty" and "fever" were not tested for sars-cov-2 earlier. It is very unlikely with patient delay from this point since the symptoms were recorded in the notes. These cases should be manually evaluated.

The method was trained on "cardiovascular, pulmonary, and metabolic diseases and phenotypes", so not gastro-intestinal? How were symptoms like diarrhea validated?

Still confusing to me that "negative sentiment classification" can have recall. I don't understand this, are there true positives in the negative classification? If so why?

---

## [Author Response]

Essential revisions:1) The cohort itself is not well-defined and should be better described (see below).

We now describe the cohorts in better detail in terms of demographics and comorbidities, as discussed below in more detail.

2) Materials and methods section should be written more clearly (see below).

Based on this feedback, we now elaborated our Materials and methods section, as discussed below in more detail.

3) Remove RNA seq data and treatment figures and discussion about these topics. It is distractive and does not make the manuscript stronger.

We accept this suggestion and have now removed figures and discussion about the RNAseq data and treatment.

Cohort description:How many patients had an admission 7 days before getting a swab taken?

This was computed and the number of patients who were admitted 7 days or more prior to the PCR testing date are shown below.

COVIDpos : 11 (0.5%, n = 2,317)

COVIDneg : 196 (0.3%, n = 74,850)

How many individuals do the authors have a full week of information on prior to the PCR?

This was computed and the number of patients who have at least one note in all 7 days prior to PCR testing are shown below.

COVIDpos : 6 patients (0.3%)

COVIDneg : 372 patients (0.5%)

Why was there a delay in PCR testing of these patients?

This inherent delay is caused by that fact that both the patient and physician must consider the symptoms enough to warrant PCR testing. More specifically, the patient must first consider the symptoms serious enough to visit the clinic and the physician must then determine that the symptoms present a possibility of COVID infection to order the test. We hope that the results presented in this manuscript, as well as more widely available testing, will help to shorten this delay in the future.

Who are the 30,000 patients, and why were they admitted to a hospital? Or were they outpatients? The setting and patient population needs to be described.

Expanding on the hospitalization counts 7 days or more prior to PCR testing (shown above), we have also computed the number of patients who were admitted between day -7 and day 0. The remainder of the patients were outpatients.

COVIDpos : 135 patients (5.8%)

COVIDneg : 5,981 patients (8.0%)

Completely basic information like sex and age are missing.

We have provided details regarding the age and sex of each cohort in the main text and present them below.

COVIDpos

Age: Mean: 41.9 Std: 19.1

Sex: M: 50.6% F: 49.4%

COVIDneg

Age: Mean: 50.7 Std: 21.4

Sex: M: 43% F: 57%

The authors report that they use 15.8 million clinical notes from 30,494 patients, so about 500 notes/patient. Are these from the 7-day period, so more than 70 notes written per day? Or from a longer non-disclosed time-span?

We agree that the number of notes presented could be a point of confusion, as the numbers presented were the total number of notes for the entire medical history of these patients at the Mayo Clinic, not just day -7 to day 0. To prevent confusion for readers, we have removed the note counts from the manuscript and instead highlight the number of patients used for this analysis.

Materials and methods section:In this study patients are analyzed until date of diagnosis (test), it seems to be an analysis of when in the natural course one chooses to test? As it is formulated in the manuscript, "temporal patterns" first of all indicate, that the population converge towards day of test, so patients "progress towards same phenotype" and it is unclear how this relates to COVID-19 progression?

We agree with the reviewer that synchronization is integral to the analyses described in this manuscript. As described previously in this rebuttal, there is an inherent delay between when COVID-19 infection occurs and when the patient and physician deem testing necessary based on the symptoms present. This delay could be influenced by how well-informed the public is regarding COVID-19 signs and symptoms, the availability of PCR testing, and the standardization of hospital protocols for which patients get tested. Each of these factors would be limited at the beginning of a pandemic, but improve over time. As is the case with real-world data, there will be outliers, but increasing the size of the COVIDpos cohort 4-fold has increased the statistical significance of many of the signals presented herein. The fact that many of the COVIDpos patients progress toward the same set of symptoms demonstrates that synchronization using the PCR testing date is apt for such analysis. By understanding the temporal progression of symptoms prior to PCR testing, we aim to reduce the delay between infection and testing in the future.

What is the link between the analysis and the references to chloroquine/hydroxychloroquine?

We agree that these references were tangential and have now removed the references to chloroquine/hydroxychloroquine.

The presentation in Materials and methods, also has many unclear aspects. For example, what was the output from also curating disease and medication? It seems, that only symptoms are presented in the manuscript?

We agree that including medications in the Materials and methods was unrelated to the analyses presented and have now removed medications from the manuscript.

How do symptom categories and phenotypes differ?

We have clarified the distinction between symptoms and phenotypes throughout the manuscript. All analyses described were performed using COVID-19 symptoms. Phenotypes, i.e. other diseases/conditions, were only present in the 18,490 sentences used for training the initial SciBERT model, which was later supplemented with 3,188 sentences containing COVID-19 symptoms. Thus, “phenotype” is only used in the Materials and methods for the description of this model.

The iteration for optimization of the model seems a little unclear and how were the 18,500 sentences in the test set selected?

We have clarified how the model was optimized in the Materials and methods section. As we now describe, the 18,490 sentences selected for the test set were randomly selected from a pool of sentences containing 250 different cardiovascular, pulmonary, and metabolic diseases and phenotypes.

What was the indication for COVID-testing in these patients? Were they all hospitalized for different conditions? Were all 30,494 under suspect for COVID-19 or were some tested simply because they were hospitalized in the period of the pandemic (i.e. routine/screening test)? And what were the diagnoses of the COVIDneg cases?

We have included hospitalization information in the responses above and in the main text of the manuscript. Additionally, we have included the most frequent ICD10 diagnosis codes for both cohorts between day -7 and day 0 in the Supplementary information.

Were there notes on all patients from index -7 to 0 as mentioned above?

Yes, all patients in both cohorts had at least one note between day -7 and day 0.

What are the demographics of these patients?

We have provided details regarding the age and sex of each cohort in the main text and present them below.

COVIDpos

Age: Mean: 41.9 Std: 19.1

Sex: M: 50.6% F: 49.4%

COVIDneg

Age: Mean: 50.7 Std: 21.4

Sex: M: 43% F: 57%

And were symptoms handled as chronic or temporary conditions?

No distinction was made between chronic and acute symptoms. We are currently developing models to extract the temporal context around a symptom, but here the model only classifies a symptom as Yes (confirmed), No (ruled out), Maybe (possibility of), or Other (alternate context).

Why was altered or diminished sense of taste and smell (anosmia/dysgeusia) included in Results despite a classification performance of 64.4%?

Our initial analysis was run using a model trained on 18,490 sentences containing 250 different cardiovascular, pulmonary, and metabolic diseases and phenotypes. After directly examining sentences containing COVID-19 symptoms, we came to the conclusion that this training data did not adequately capture the context in which these symptoms were described. Thus, we manually labeled 3,188 sentences containing COVID-19 symptoms and used these in combination with the 18,490 sentences to train a new model. This new model was used for all of these analyses described herein, and resulted in an overall accuracy of 85.6% of altered or diminished sense of taste and smell.

Not sure why there are two F scores. Why calculate F scores on sentences labeled as "not present" – how is recall not undefined in such a calculation?

We apologize for the confusing wording used previously. We have revised the text and included tables for all models, which contain the F1-Scores for each classification.

How were the sentences in step 2 and 3 chosen?

We have revised the Materials and methods regarding model development to more accurately reflect the workflow. As we now describe, 18,490 sentences were selected randomly from a pool of sentences containing 250 different cardiovascular, pulmonary, and metabolic diseases and phenotypes. Similarly, 3,188 sentences were selected randomly from a pool of sentences containing COVID-19 related symptoms.

Why is the sum of the true positive rate and the false positive rate not 100%?

We believe the reviewer meant to ask why the true positive rate and false negative rate do not equal 100%, as they should. This was a labeling error, which has been fixed in Supplementary file 1H and now shows the true positive rate and false positive rate.

Confidence intervals would help the interpretation of the data. It would be great if the authors would provide number of tests or the significance level to help interpret the Benjamini-Hochberg correction.

We have added confidence intervals to both Table 1 and Supplementary file 1D. We have also clarified that the Benjamini-Hochberg correction was applied on 351 Fisher test p-values.

2) The authors state that the platform utilizes state-of-the-art transformer neural networks. But they used BERT (original version) indeed. Bert is not the state-of-the-art transformer model. XLNet and RoBERTa are the state-of-the-art models. For name entity recognition of clinical note data, there have been some specific BERT-based variations or pre-trained models, such as BioBERT.The authors did not provide details of the implementation of BERT configuration, like how many layers, heads? How they train BERT, like how many epochs? what is the optimizer setting? Did they use a pre-train (on which corpus) BERT? etc. All these details need to be provided for reduplication. My suggestion is better to provide several examples regarding what the input looks like ,i.e., the unstructured clinical notes, and corresponding structured output, so that readers can understand how powerful the model is.

We have addressed these concerns in the Materials and methods and Figure 1—figure supplement 1-2, showing the SciBERT architecture, training configuration, and example sentences from the unstructured text of clinical notes with their classifications.

3) The dataset split (90%/10%) is not usual. "70%/10%/20%" or "60%/10%/30%" are more common for fair evaluation in deep learning (the middle one represents the validation set). Could the authors provide the reason why they split dataset in this way or provide a reference that applied this strategy?

At the time of training, since we had limited tagged examples, we wanted to maximize the size of the training and test sets. Thus, when choosing our models, we used the test set accuracy to determine the optimal number of epochs to stop at.

4) In the Results section, the first paragraph is about details of data collection, it would be more appropriate in part of the Materials and methods, and the final paragraph would be more appropriate in the Discussion. In Table 1, the proportion columns would be better to combine with the patients-count columns.

We have revised the first paragraph of the Results section to focus on characterization of the COVIDpos and COVIDneg cohorts. We have also revised the final paragraph to better summarize the Results prior to the Discussion. Finally, we have combined the columns for the patient counts and proportions as suggested by the reviewer.

5) In terms of the Discussion, it would be important to emphasize and discuss the main findings. "By contrasting EHR-derived phenotypes of COVID pos (n = 635) versus COVID neg (n = 29,859) patients during the week preceding PCR testing, we identify anosmia/dysgeusia (37.4-fold), myalgia/arthralgia (2.6-fold), diarrhea (2.2-fold), fever/chills (2.1-fold), respiratory difficulty (1.9-fold), and cough (1.8-fold) as significantly amplified in COVID pos patients" This statement in the Abstract should be the key finding, but the authors did not emphasize the statistical method and make discussion properly.

We have revised the Discussion to focus on the implications of identifying an early symptom signature for COVID-19 prior to testing.

6) The pairwise analysis of phenotypes considered only the combination of 2 phenotypes. How to process the combination of multiple phenotypes?

We have also considered combinations of 3 and 4 symptoms. However, as we increase the number of symptoms, the results become more and more under-powered. Thus, we have chosen to only include those for combinations of 2 symptoms.

7) As the NLP-based entity recognition can bring errors, the following statistic analysis could be biased by these errors. The authors should emphasize this point. I would suggest the authors to use the manually curated data from the first 100 patients to perform the same analysis to see if it can generate the same results.

We have now emphasized the potential sources of error in the Materials and methods. Performance on the manually curated data is represented in Supplementary file 1H.

[Editors' note: further revisions were suggested prior to acceptance, as described below.]

You have definitely made a number of changes that have improved the readability and consistency of the manuscript. Although the paper essentially only presents already known symptoms of the disease (e.g https://doi.org/10.1136/bmj.m1996), it is now clearer that the paper aims to characterize when and how long prior to PCR-diagnosis a symptom is overrepresented.However, one reviewer comments that there are still many inconsistencies in the manuscript that at best makes it difficult to read but also bring uncertainty about the validity of the results. Materials and methods section paragraphs are still hard to follow to the extent that the study would be hard to reproduce.

We thank the reviewers and the editors for their feedback and address them below.

Indeed, as the editors have suggested, our study characterizes when and how long prior to PCR-diagnosis a symptom is overrepresented. We would like to highlight that there are other key differentiators too from other other studies such as the BMJ paper by Argenziano et al. This BMJ paper is a manual curation effort to comb through 1000 COVID-19 patients EHRs. Please see this statement from the paper

(https://doi.org/10.1136/bmj.m1996): “An abstraction team of 30 trained medical students from the Columbia University Vagelos College of Physicians and Surgeons who were supervised by multiple clinicians and informaticians manually abstracted data from electronic health records in chronological order by test date.”

We would like to point out that our augmented curation technology premised on neural networks is highly scalable and efficient, whereas the manual curation approach used by the above study, continues to be the current state-of-art despite the associated disadvantages of scalability. Clearly, there is a need to validate any augmented curation methods, which we have now conducted and included in the context of COVID-19 phenotypes.

The BMJ paper and others like it typically look at severe disease or critically ill patients from the ICU (emergency department, hospital wards and ICUs). Our manuscript, on the other hand, looks at the whole spectrum of COVID-19 patients, including from outpatient visits.

Finally, it is noteworthy that our manuscript contributed to flagging anosmia (loss or alteration of sense of smell) as an early indication of SARS-CoV-2 infection, despite a majority of physicians at the time of this paper’s submission not explicitly asking patients regarding a newfound sense of loss of smell. It may further be noted that our preprint on medRxiv and the subsequent conversations that the Mayo Clinic COVID-19 task force led with the state and federal governments contributed towards the CDC altering its guidelines to explicitly add altered or loss of smell or taste to its new guidelines for SARS-CoV-2 management:

(https://www.cdc.gov/coronavirus/2019-ncov/symptoms-testing/symptoms.html).

Specific comments:1) Altered smell is as mentioned not a new finding, the authors also report this in relatively small numbers (6.3%) for covid19 patients, it seems unclear how much of an impact this would have clinically. The authors should comment on how this differs from other studies. In the context Tim Spector's work using app may also be relevant.

We would be happy to address this comment. In the paper by Tim Spector and colleagues focused on self-reporting of symptoms, it is unclear how many days prior to PCR-based COVID-19 diagnosis the reporting of symptoms is available. It is possible that many of the symptoms were noted after diagnosis too.

While the study by Tim Spector and colleagues indeed does point to 65.03% of the SARS-CoV-2-positive patients experiencing a loss of smell or taste, it needs to be taken into consideration that 21.71% of the SARS-CoV-2-negative patients also reported this phenotype (https://www.nature.com/articles/s41591-020-0916-2). Hence, this only represents about a three-fold enrichment of anosmia/dysgeusia in COVIDpos over COVIDneg patients. However, our study identifies a much higher enrichment (27-fold) of anosmia/dysgeusia in COVIDpos over COVIDneg patients.

Finally, it also needs to be considered that the app explicitly asks the question of whether an altered sense of smell or taste was encountered by each participant, which is different from the real-world clinical setting, particularly during the early phases of the pandemic when anosmia/dysgeusia was not part of the CDC guidelines.

2) I could not follow the number of symptoms (26 or 27?) and how were these 26 or 27 selected? Who selected these and what was their backgrounds Why not search for overrepresented symptoms in general and include way more symptoms? Currently only suspected/known symptoms were included. "Examining the other 325 possible pairwise symptom combinations" which were the first 26 for example. In Table 1, why is respiratory failure and dysuria marked in a light grey color?

We thank the reviewer for pointing out the mistyped number of symptoms, which has been fixed in the revised version of the manuscript.

During the week preceding PCR testing, 100 COVIDpos patients’ clinical notes were examined to select the over-represented symptoms. The list of symptoms considered was examined and approved by the COVID-19 taskforce at Mayo Clinic, and physicians from multiple specialties that are actively undergoing patient care for COVID19 patients across a broad spectrum of phenotypes. Conducting further holistic phenotypic scan will be the topic of follow-up studies upon publication of this manuscript – potentially as a Research Advance to *eLife* in ~6 months, at which point the patient counts would have also significantly increased, thus amplifying the overall statistical confidence of the day-by-day enriched signals preceding COVID-19 diagnosis.

The light gray shading for the dysuria and respiratory failure rows in Table 1 was an artifact of a previous draft and has been removed in the revised manuscript. We thank the reviewer again for helping us clarify this.

3) Why is there automated entity recognition of "drugs"? Drug are not otherwise mentioned in the paper? There is still a part about scRNA-sequencing in the results, but no results about RNA-seq presented.

The reference to “scRNAseq” and “drugs” were in the context of broader discussion and introduction points. Since they appear to be distracting, we have now removed them in the revised manuscript. We thank the reviewer for pointing out.

4) Why was the method applied to the 35 million notes? When only notes from 7 days are analyzed in the study… I don't agree with deleting information about the number of notes, it does not solve the problem from the original submission. It would be nice for the reader to be able to evaluate the number of notes or words written about the included patients. Also, how many notes are from day 0? Are notes recorded at day 0 included or not, there seems to be inconsistency here throughout the manuscript. The BERT validation seems to be done on notes including day 0, whereas the actual proportion analyses seem to be excluding day 0 which is the day one must expect to have most notes concerning covid19.

The model was indeed applied to over 35 million notes at the onset of this study, which represents the entire medical record of the patients considered for this analysis. However, as the reviewer correctly points out, only 7 days prior to PCR testing date was ultimately decided to be of immediate interest from a SARS-CoV-2 diagnostic standpoint. In order to make this clear, the number of patients and associated notes on each day of the week preceding SARS-CoV-2 PCR testing has now been included explicitly in Supplementary file 1E.

Day 0 was mentioned in the table for reference and is not part of any analysis barring the model training. Obviously, day 0 presents no predictive insight as to the outcome of the SARS-CoV-2 PCR test which is also conducted on that same day, and hence is not included for any further interpretation.

5) Many basic aspects are still unclear. How many data points could one patient contribute with? Could one patient tested multiple times contribute more than once? If a patient was tested negative and then positive, how was this handled?

Each patient is counted only once. Once they have a positive SARS-COV-2 PCR test, they are considered COVIDpos. If a patient were to test negative and then positive subsequently, that day of positive PCR testing is considered day 0 for that patient. We now mention this clearly in the Materials and methods section of the current manuscript.

“*Each patient is counted only once. Once they have a positive SARS-COV-2 PCR test, they are considered COVIDpos. If a patient were to test negative and then positive subsequently, that day of positive PCR testing is considered day 0 for that patient.*”

6) It a considerable limitation that there is no separation between chronic and acute symptoms. If the condition is chronic it wouldn't help at all in the diagnosis, only that this population for some reason is more prone to be covid19 positive.

Based on currently available literature and current CDC guidelines (https://www.cdc.gov/coronavirus/2019-ncov/need-extra-precautions/people-withmedical-conditions.html), there are indeed multiple specific underlying risk factors associated with COVID-19 diagnosis (e.g. diabetes, hypertension, obesity, chronic kidney disease, COPD, immunocompromised patients, serious heart conditions, sickle cell disease). However, to our knowledge, there are no known associations to chronic versions of symptoms described in our manuscript (e.g. “chronic cough”, “chronic diarrhea” or “chronic anosmia”). Furthermore, the day-to-day variability of the phenotypes also clearly argue in factor of the considered phenotypes not being the result of a generic chronic underlying condition but rather that of SARS-CoV-2 infection onset and early COVID-19 progression. Finally, at the level of the population, the signals that we present are fairly robust given the large cohorts of COVIDpos and COVIDneg patients considered.

We do intend to continue monitoring and reporting the updated statistics, as noted above, via potentially a Research Advance to *eLife* in ~6 months, at which point the patient counts would have also significantly increased and provide an extended time-line to examine chronic aspects associated with COVID-19 as well.

7) It still remains unclear why patients with obvious covid19 symptoms like "respiratory difficulty" and "fever" were not tested for sars-cov-2 earlier. It is very unlikely with patient delay from this point since the symptoms were recorded in the notes. These cases should be manually evaluated.

In this “real world evidence” based study, there are multiple factors that could contribute to the time before the test. The early stages of the pandemic did not necessarily involve patients getting PCR tests at the very first semblance of many of these common phenotypes, and furthermore there was the limited availability of the PCR tests themselves during such early days of the pandemic. During the further stages of the pandemic, there has been a large push to self-isolate upon symptom manifestation, which might have delayed patients from coming into the hospital to get tested, despite discussions either over tele-medicine or in-person visits with physicians that didn’t culminate in a PCR test at the first juncture.

Furthermore, it is quite possible that negative PCR test results were indeed produced in some patients during the week preceding the eventual first positive PCR test result. It is also important to note that, many of these patients in whom PCR may come our negative during early days of the infection, may well present with mild symptomatology such as anosmia. Indeed, there is indeed emerging evidence in the scientific literature that does show that PCR testing results being positive may be delayed depending on the nature and extent of the nasopharyngeal swab conducted, the ability to gather sufficient replication-competent virions in such swab specimen, as well as other factors remaining to be understood that constitute some of the caveats of false negative PCR testing. Regardless, it is a fact that PCR testing continues to be the gold standard for positive SARS-CoV-2 diagnosis till date and is further unlikely to be substituted anytime in the near future.

In the manuscript, we have the following text that clarifies these points:

"There are a few caveats that must be considered when relying solely on EHR inference to track symptoms preceding the PCR testing date. In addition to concerns regarding testing accuracy, there is an inherent delay in PCR testing, which arises because both the patient and physician must decide the symptoms warrant PCR testing. More specifically, to be tested, the patient must first consider the symptoms serious enough to visit the clinic and then the physician must determine the symptoms present a possibility of COVID infection. The length of this delay could also be influenced by how well-informed the public is of COVID-19 signs and symptoms, the availability of PCR testing, and the hospital protocols used to determine which patients get tested. Each of these factors would be absent or limited at the beginning of a pandemic but would increase or improve over time. However, this makes synchronization across patients difficult because the delay between symptom onset and PCR testing changes over time. For example, patients infected early in the pandemic would be less inclined to visit the clinic with mild symptoms, while those infected later have more information and more cause to get tested earlier. Similarly, in the early stages of the COVID-19 pandemic when PCR testing was limited, physicians were forced to reserve tests for more severe cases or for those who were in direct contact with a COVID*pos* individual, whereas now PCR testing is more widespread. In each case, the delay between symptom onset and PCR testing would be expected to change over time for a given patient population."

The method was trained on "cardiovascular, pulmonary, and metabolic diseases and phenotypes", so not gastro-intestinal? How were symptoms like diarrhea validated?

The original diagnosis model was trained using cardiovascular, pulmonary, and metabolic diseases/phenotypes (Supplementary file 1G). Because this model was not trained on COVID-related symptoms, e.g. gastrointestinal symptoms, additional sentences were manually labeled containing COVID-19 symptoms, such as diarrhea. To make a new, more generalizable model, these sentences were included in the training and validation of a new model composed of sentences from the former model and these additional COVID-related symptoms (Supplementary file 1H). Validation of this model was performed on 4001 sentences containing COVID-related symptoms (Supplementary file 1I).

In fact, a custom model for each organ/disease is not the ideal way to develop the neural network based augmented curation system. The authors are of the belief that a model that is trained on a broader range of phenotypes and indications would generalize better to many diseases and phenotypes that the model hasn’t actually encountered in the past. These views are being explored through a detailed follow-up study comparing models trained on specific corpora of knowledge as opposed to a “core corpus” that includes multiple diverse indications not biased/defined by current siloes.

Still confusing to me that "negative sentiment classification" can have recall. I don't understand this, are there true positives in the negative classification? If so why?

Please consider the following example of a hypothetical sentence from a provider’s clinical note, “The patient does not have diarrhea today”. This sentence will be classified by our model as the result: “NO”, for the symptom: “diarrhea”. Such a negative sentiment would be regarded by our model as a “true positive” for the lack of diarrhea. Thus, a “negative sentiment classification” can have recall computed from these true positives.